# Neutralisation of the Immunoglobulin-Cleaving Activity of *Streptococcus equi* Subspecies *equi* IdeE by Blood Sera from Ponies Vaccinated with a Multicomponent Protein Vaccine

**DOI:** 10.3390/vaccines13101061

**Published:** 2025-10-17

**Authors:** Francesco Righetti, Karina Hentrich, Margareta Flock, Sara Frosth, Karin Jacobsson, Joakim Bjerketorp, Anuj Pathak, Noela Ido, Birgitta Henriques-Normark, Lars Frykberg, Romain Paillot, Bengt Guss, Tim Wood, Jan-Ingmar Flock, Andrew Stephen Waller

**Affiliations:** 1Department of Microbiology, Tumor and Cell Biology, Karolinska Institutet, 171 77 Stockholm, Sweden; 2Clinical Microbiology, Karolinska University Hospital, 171 76 Stockholm, Sweden; 3Department of Biomedical Science and Veterinary Public Health, Swedish University of Agricultural Sciences, 750 07 Uppsala, Sweden; sara.frosth@slu.se (S.F.);; 4Intervacc AB, 129 22 Stockholm, Sweden

**Keywords:** IgG endopeptidase, neutralisation, strangles, vaccination, horse

## Abstract

**Background:** *Streptococcus equi* subspecies *equi* (*S. equi*) is the cause of strangles, one of the most prevalent diseases of horses worldwide. The disease is characterised by fever and the formation of abscesses in the lymph nodes of the head and neck, which can restrict the airway. A multicomponent subunit vaccine, Strangvac, has been shown to effectively reduce clinical signs of strangles and to reduce its incidence. **Objective:** The aim of this study was to determine the immune response against the immunoglobulin-cleaving endopeptidase IdeE, a key protective component within the vaccine and the ability of antibodies to neutralize the proteolytic activity of IdeE. **Methods:** An in vitro assay was developed to measure the functional inhibition of recombinant IdeE by horse sera pre- and post-vaccination. The IdeE-neutralising titres were compared to the corresponding IdeE-specific antibody titres measured by iELISA (indirect Enzyme-Linked Immunosorbent Assay). **Results:** A significant IdeE-specific antibody response in blood serum collected from ponies was induced after Strangvac vaccinations. Concomitantly, significant increases in the neutralising activity of IdeE occurred, persisting for at least 12 months post-second vaccination. IdeE-neutralising activity was further increased significantly after a third vaccination, even when the third dose was administered 12 months after the second dose, demonstrating that immunological memory to the vaccine persisted for 12 months. There was a significant correlation between the IdeE-neutralising activity of blood sera and the level of IdeE-specific antibodies. **Conclusions:** These data provide insights into one potential mechanism by which this vaccine protects Equids against or during *S. equi* infection.

## 1. Introduction

*Streptococcus equi* subspecies *equi* is one of the most prevalent infectious diseases of horses worldwide [1,2,3,4]. The disease is characterised by pyrexia, pharyngitis, cough while eating, dysphagia, inappetence, mucopurulent nasal discharge and the development of abscesses in the lymph nodes of the head and neck [1,5]. An important aspect of the infection is that it is highly contagious. As the disease progresses, abscesses in the retropharyngeal lymph nodes rupture and drain into the guttural pouches and then into the environment via the nasopharynx. Incomplete drainage of abscess material from the guttural pouches or incomplete clearing of a subclinical infection results in a proportion of convalescent animals becoming persistently infected with *S. equi*. These outwardly healthy ‘carriers’ intermittently shed *S. equi* into the environment, where the bacterium can be taken up by naïve animals, triggering new outbreaks of disease [6,7,8].

Persistently infected carriers and affected horses with severe signs of upper airway obstruction, dysphagia, purpura haemorrhagica and bastard strangles often require treatment with antibiotics over extended periods [1,8]. Despite the risk of the development of anti-microbial resistance (AMR), antibiotics are also sometimes used to prevent the formation of abscesses post-exposure to *S. equi* [9]. However, once abscesses have formed, antibiotics often fail to penetrate sufficiently to kill *S. equi* [1]. Furthermore, mutations in the *pbp2X* gene, encoding an important penicillin-binding protein of *S. equi*, have been identified in isolates recovered from horses in Europe and the USA [10]. Therefore, the use of quarantine, biosecurity, diagnostic testing and vaccination will play increasingly important roles in the prevention of *S. equi* infection thereby minimising the risk of the development of AMR [11].

A recombinant fusion protein vaccine (Strangvac, Intervacc AB, Stockholm, Sweden) resulted in significantly lower clinical signs of infection, such as fever, lymph node score, coughing, demeanour score, feeding score and inflammation two weeks after the second vaccination, and 31% (5 of 16) of ponies were fully protected lacking any sign of infection. After a third vaccination, 94% (15 of 16) of ponies were fully protected from experimental challenge that induced disease in all control animals [12]. Protection after only two vaccinations was shown to remain for at least two months but may persist for much longer since significantly elevated antibody levels were demonstrated after twelve months [12]. Experience of efficacy and safety from the use of Strangvac in the field has been documented [13]. The vaccination generated antibody responses against the endopeptidase IdeE (SEQ0999), against Eq85, which encompassed two *S. equi* proteins (Eq8 [SEQ0402], Eq5 [SEQ0256]), and against CCE, which was comprised of five *S. equi* proteins (CNE [SEQ0935], SclC [SEQ2101], SclF [SEQ0855], SclI [SEQ1817] and EAG [SEQ0721]) [12]. Although this might reflect an antibody response against each individual component of the fusion proteins (J-I. Flock, personal communication, 23 February 2022), the relative functional contribution of each protein, and particularly the non-fusion protein IdeE, to the immune response is uncharacterised.

Whereas all proteins in Eq85 and CCE are sortase-processed surface proteins [14], IdeE is a secreted immunoglobulin G (IgG)-specific endopeptidase [15]. There is increasing evidence that immune responses towards immunoglobulin endopeptidases are important for protection against *S. equi* subspecies *equi.* The vaccination of mice with IdeE or IdeE2, another IgG endopeptidase of *S. equi*, induced protection against experimental infection with *S. equi* subspecies *equi* [16]. Furthermore, an addition of IdeE and IdeE2 to a prototype vaccine (Pentavac) containing five components resulting in a seven-component vaccine (Septavac) increased the level of protection that was conferred against experimental challenge of ponies with *S. equi* [17]. However, in a subsequent study using a vaccine where IdeE2 was omitted but IdeE was retained, protection from disease in ponies was not reduced, providing strong evidence of the contributory effect of IdeE to the protective effects of Strangvac [14].

Here we report the development of an in vitro assay to measure the ability of sera collected pre- and post-vaccination of ponies with Strangvac to neutralise the activity of recombinant IdeE

## 2. Materials and Methods

### 2.1. Study Design

The blood serum samples that were analysed in this study were collected from 28 Welsh mountain ponies that participated in two vaccination experiments reported previously [12]. For consistency and alignment, the same nomenclature used for the two published vaccination experiments (i.e., Experiment [Exp.] I and Exp. IV) will be adopted in this study and are depicted in Figure 1. Exp. I had three arms where V3 was given at three, six or twelve months after V2.

These experiments were conducted under the auspices of a Home Office Project License according to the Animal Scientific Procedures Act 1986 and following ethical review and approval by the Animal Health Trust’s Animal Welfare and Ethical Review Body (Research Project Proposal (RPP) 01_08; Approved in May 2008). Further animal welfare information for these studies is included in Appendix A.

The vaccine used in these studies (Strangvac, Intervacc AB, Stockholm, Sweden) contained recombinant *S. equi* protein antigens derived from the BAPS-2 *S. equi* strain 1866 (*Se*1866) [12,18]. The soluble antigenic components of the vaccine were produced in *E. coli* strain BL21 and purified as previously described [14]. Twelve ponies in Exp. I (two males and ten females, aged 10 to 26 months) and sixteen ponies in Exp IV (ten males and six females, aged 4 to 5 months) were vaccinated by intramuscular administration of the vaccine containing 50 µg of each antigen and 326 µg of Matrix V (Novavax, Gaithersburg, MD, USA) in two mL total volume, according to the schedule shown in Figure 1. An additional group of 16 ponies in Exp. IV (eight males and eight females aged 4 to 5 months) acted as controls and were vaccinated with a placebo containing only adjuvant. Samples from eight animals from the vaccinated and the placebo groups, respectively, were included in this study.

Prior to entry into the study, ponies were examined by a veterinarian and tested to confirm their infection-free status, lack of recent exposure to *S. equi* and overall health. All were maintained at pasture for the duration of the vaccination phases. Ponies in Exp. IV were challenged using a two ml culture of *S. equi* strain Se4047 containing 5 × 10^7^ cfu sprayed into each nostril (1 × 10^8^ cfu per pony in total) at two weeks post-third vaccination (day 133) [12].

### 2.2. Collection of Blood Samples

Blood samples for this study were drawn from the jugular vein of ponies pre-first vaccination (day 0 in Exp. I and day-1 in Exp. IV), pre-third vaccination (days 119 [group 1], 210 [group 2] and 392 [group 3] in Exp. I and day 118 in Exp. IV) and two weeks post-third vaccination (days 135 [group 1], 226 [group 2] and 408 [group 3] in Exp. I and day 132 in Exp. IV) for quantification of IdeE responses and IdeE-specific antibodies (or IgG) responses. Blood samples were allowed to clot at room temperature for two hours. The serum was removed and stored at −20 °C. Sera were then thawed at room temperature and subsequently analysed after the vaccination studies were concluded [12]. Samples from all 12 vaccinated ponies in Exp. I and from 8 (randomly picked) of the 16 ponies in each of the placebo and vaccinated groups in Exp. IV were included in the current study.

### 2.3. Quantification of IdeE Neutralisation

Recombinant IdeE (0.25 ng, final concentration of 0.01 μg/mL) was prepared in phosphate buffered saline (PBS) supplemented with 1% inactivated naïve horse serum and 1 mM dithiothreitol (DTT), to a volume of 17.5 μL. Five microlitres of five-fold serial dilutions, from 10% of the tested sera, were added to obtain final serum dilutions of 1:10, 1:50, 1:250, 1:1250 and 1:6250. Reactions were incubated for one hour at 37 °C to permit binding to IdeE. Human IgG1 (2.5 μg; Millipore Sigma, Burlington, MA, USA) was added and the reaction was incubated overnight (16 h) at 37 °C as described previously [15]. Human IgG1 was used as it was observed that human IgGs are efficiently cleaved by *S. equi* IdeE [15,16]. The lower hinge region of human IgG1 is 100% identical to horse IgG1 and IgG3, including the LGG motif that is targeted by the enzyme. Human IgG1 is readily available and allows the specific detection of the input substrate, differentiating from the horse antibodies in the test sera. To stop the reaction, 25 μL of two-fold protein buffer containing sodium dodecyl sulphate (SDS) and β-mercaptoethanol was added.

Samples were heated at 95 °C for 15 min, and 5 μL were analysed by electrophoresis at 160 Volts for one hour on a 4–12% Bis-Tris gel (Thermo Fisher Scientific, Waltham, MA, USA). Proteins were transferred to Polyvinylidene fluoride (PVDF) membranes using the Trans-Blot Turbo system (Bio-Rad, Hercules, CA, USA). Membranes were blocked in 5% milk dissolved in PBS + 0.1% PBS Tween (PBST) for 1 h at room temperature and incubated 1 h at room temperature with anti-human IgG fragment crystallizable (FC) horseradish peroxidase (HRP)-conjugated antibody (Thermo Fisher Scientific, Waltham, MA, USA), diluted 1:20,000 in PBST. After washing the membranes three times for five minutes with PBST, they were treated with Amersham enhanced chemiluminescent (ECL) reagents (GE Healthcare Life Sciences, Chicago, IL, USA) and visualized by GelDoc XRS+ (Bio-Rad). All the Western blot images are shown in Appendix A, for Exp. I and IV, respectively. All the original blots are collected in Appendix A. Signals were quantified with the software Image Lab (Bio-Rad, version 6.0.1) and the ratio of the intensity of the uncleaved (~55 kDa) and cleaved (~25 kDa) products was calculated. An assay cut-off for the ratio of uncleaved/cleaved of >1.0 was applied as a measure of IdeE neutralisation. For the purposes of data analysis, a titre of two was applied to those serum samples where there was no IdeE-neutralising activity at a 10-fold dilution (i.e., neutralising activity < 10). The IdeE neutralisation titres are listed in Appendix A. Appendix A contains the measured densitometry of the signals detected in the Western blots and the calculation of the IdeE neutralisation titres.

### 2.4. Quantification of the Antibody Response to Vaccination

Serum samples were also used to measure IdeE-specific Ab (antibodies; or IgG) responses by conventional IdeE iELISA, performed as previously described [12,17]. Samples were diluted two-fold serially and the Log_10_ value of the dilution required to obtain an absorbance value below a cut off threshold of 1.5 was determined. The IdeE-specific antibody titres are listed in Appendix A.

### 2.5. Statistical Methods

Normality was tested with the Shapiro–Wilk test. Data were found to be non-normally distributed; the non-parametric Friedman test (for repeated measures) and Kruskal-Wallis test (for independent measures) were used as primary analyses. Post hoc pairwise comparisons were performed using the Nemeney test (for Friedman) and Bonferroni-corrected tests (for Kruskal–Wallis), with significance set at *p* < 0.05. Statistical differences in IdeE-neutralising titres were analysed using a one-tailed Mann Whitney *U* test or a paired Wilcoxon signed rank test, with significance set at *p* < 0.05. One-tailed analysis was used when comparing pre- and post-vaccination titres or placebo and vaccinates, since vaccination only increased (never decreased) antibody titres and concomitantly the neutralisation titre (directional assumption with H_1_: pre-V < post-V). The Spearman’s Rank correlation test was used to determine if there was a significant relationship between the IdeE-specific antibodies in sera as measured by iELISA and the IdeE-neutralising titre of sera [19], with significance set at *p* < 0.05.

## 3. Results

### 3.1. Recombinant IdeE Cleaved Human IgG1

Overnight incubation of 2.5 µg human IgG1 in the presence of 0.25 ng of recombinant IdeE generated a cleaved IgG1 product that was detected by the anti-human IgG FC HRP-conjugated antibody (Figure 2).

### 3.2. Duration of IdeE-Neutralising Activity After V2

Western blots showing the neutralising activity of blood serum samples in Exp. I are shown in Appendix A, and IdeE-neutralising titres are reported in Appendix A. Blood serum samples taken pre-first vaccination (day 0) had little or no neutralising activity against the IgG1-cleaving action of IdeE (median < 10; Figure 3), with only one outlier with a neutralising titre of 50. Serum samples taken after V2 but immediately prior to the third vaccination (day 119 for group 1; day 210 for group 2 and day 392 for group 3) showed a significant IdeE-neutralising activity when compared with pre-first vaccination (*p* = 0.00079). Pre-vaccination IdeE-neutralising titre (day 0) were significantly lower when compared with group 1 pre-V3 3 months (day 119; median 10; *p* = 0.0158), group 2 pre-V3 6 months (day 210; median 10; *p* = 0.0026) and group 3 pre-V3 12 months (day 392; median 50; *p* = 0.0012) (Figure 3). Pre-V3 at 3, 6 and 12 were not different from each other, showing no significant difference after Bonferroni correction.

### 3.3. IdeE-Neutralising Activity After V3 Boost Immunisation

IdeE-neutralising titres measured 2 weeks after V3 (combined data, irrespective of the study group or time since V2) were significantly higher (median = 50; *p* < 0.013) when compared with titres measured immediately before V3 (median = 50) (Figure 4A).

IdeE-neutralising titre measured in group 3 at 2 weeks after V3 (12 months after V2) were also significantly higher (*p* = 0.029) than titres measured at pre-first vaccination (day 0) (Figure 4B), demonstrating an effective boost of IdeE-neutralising activity at the longest V2–V3 interval tested (i.e., 12 months after V2).

### 3.4. Comparison of IdeE-Neutralising Responses of Vaccinated and Placebo-Vaccinated Ponies

Western blots showing the neutralising activity of blood serum samples in Exp. IV are shown in Appendix A, and IdeE-neutralising titres are reported in Appendix A.

The IdeE-neutralising activity of sera from vaccinated ponies increased from a median titre of six at day 1 to one hundred fifty at day 118 (*p* < 0.024) and seven hundred fifty at day 132 (*p* < 0.0021, Figure 5A). IdeE neutralisation was not significantly different between day 118 and 132 (*p* = 0.35). These results support Exp. I results reported in Section 3.2 and Section 3.3.

The median IdeE-neutralising titres of sera from ponies in the placebo and vaccinated groups pre-first vaccination was not significantly different (*p* = 0.47). However, the median titres in the vaccinated group were significantly elevated at both day 118 (*p* = 0.024) and day 132 (*p* = 0.021) when compared with the placebo group measured at the same time point (Figure 5B).

### 3.5. Correlation Between IdeE-Neutralising Responses and the Titre of IdeE-Specific Antibodies in Sera

The IdeE-neutralising titre was modelled against the IdeE-specific Ab titre (Figure 6A, IdeE-neutralising titres and IdeE-specific antibody titres are reported in Appendix A). Results of the Spearman correlation test indicated that there was a significant strong positive relationship between IdeE-neutralising titre and IdeE-specific Ab titre (r(82) = 0.673, *p* < 0.001) [19], with the variation in IdeE-specific Ab titre explaining 45% of the variation in IdeE-neutralising titre (r^2^ = 0.454). Immediately before V3 (pre-V3; Figure 6B), a significant strong positive relationship (r(18) = 0.579, *p* = 0.0075) was also present. However, after V3, no significant correlation (*p* = 0.97) was measured as most IdeE-specific antibodies titres values reached a peak plateau level with little individual variation. In Exp. IV there was a correlation between neutralization titre and clinical outcome when both vaccinated and placebo groups were included. However, no such correlation could be demonstrated when including samples only from the vaccinated animals due to the small sample size and the fact that none of the vaccinated animals developed clinical signs of strangles.

## 4. Discussion

Approximately 75% of foals that recovered from natural strangles were immune to experimental challenge with *S. equi* 6 months after recovery [1,20]. Resistance to *S. equi* infection was associated with high levels of serum SeM-specific antibody levels [21]. However, the vaccine examined in the present study, Strangvac, does not contain SeM and instead is comprised of eight recombinant *S. equi* proteins, raising questions regarding the importance of each of these antigenic components to protective immunity [12].

This study demonstrated that a significant IdeE-specific antibody response was present in blood serum collected from ponies at 3, 6 and 12 months post-second vaccination and that this increased following the administration of a third dose. Concomitantly, significant levels of neutralising activity of IdeE were measured post-second and third vaccinations. These results were reflected in several observations from both Exp. I and IV.

There are several likely reasons IdeE may play an important role in the protective immune response to *S. equi* subspecies *equi.* First, IdeE cleaves IgG in vitro in a similar manner to IdeS, an important virulence protein of *S. pyogenes* [15,16]. Given that IgG plays a pivotal role in the adaptive immune system protecting equids from invading micro-organisms [16], its cleavage and neutralisation by IdeE may significantly hamper the immune response to invading *S. equi*. IdeE shares 69% amino acid identity with IdeS (also known as Mac) [22,23,24], which cleaves IgG in the hinge region to reduce the ability of phagocytic cells to recognise and kill *S. pyogenes* [23,25]. Human patients that had pharyngotonsillitis and erysipelas developed antibodies to IdeS, which neutralised its enzymatic activity, suggesting that this response may play an important role in recovery from *S. pyogenes* infection in humans [26]. Similarly, some ponies in this study had antibodies against IdeE and neutralization activity before vaccination, presumably due to previous exposure to *S. zooepidemicus*, which is endemic in populations of horses and encodes a homologue of IdeE. Moreover, IdeS has been tested in combination with other virulence factors as a vaccine candidate against *S. pyogenes* in mice [27,28]. Vaccinated mice produced antibodies against IdeS and mouse sera were able to inhibit the activity of recombinant IdeS, similarly to what was found in this study. *S. suis* also encodes a protein (Ide*_Ssuis_*) with homology to IdeE (27% sequence identity over the enzymatic region), but this enzyme cleaves IgM instead of IgGs. Vaccination with recombinant Ide*_Ssuis_* has been shown to protect pigs from a challenge with serotype 9 *S. suis* [29]. Therefore, the accumulating evidence highlights the importance of including antibody-degrading virulence factors as vaccine targets. It is also important to note that IdeE has been shown to bind equine neutrophils and to inhibit their phagocytic activity against *S. equi* in vitro [30]. Disruption of the C3-CD11b/CD18 interaction was suggested as another immune pathway targeted by IdeE to evade the immune response of equids to invading *S. equi*.

Second, when recombinant IdeE is used as an antigen in a vaccine, protection against challenge with *S. equi* was induced. For example, vaccination with recombinant IdeE induced protection against *S. equi* disease in a mouse infection model [16]. Vaccination of seven Welsh mountain ponies with three doses of a seven-component vaccine, Septavacc, containing recombinant EAG, CNE, SclC, Eq5, Eq8, IdeE and IdeE2 protected six (86%) from developing lymph node abscesses following experimental infection with *S. equi* strain *Se*4047 [17]. In contrast, only one of seven ponies (14%) was protected when a five-component vaccine (Pentavacc, Intervacc AB), which lacked IdeE and IdeE2, was used [17]. IdeE2 more efficiently cleaved equine IgG than IdeE [15,16]. However, further studies suggested that the addition of IdeE2 in a nine-component vaccine that also contained IdeE, conferred no better protection to ponies against *S. equi* infection than an eight-component vaccine, which lacked IdeE2 [14]. Therefore, the generation of an immune response against IdeE and the neutralisation of this virulence mechanism is likely to play an important role in the protection that is afforded by vaccination.

Levels of IdeE neutralisation increased post-third vaccination regardless of whether this was performed at 3, 6 or 12 months post-second vaccination. These data indicate that an immune-memory response was maintained and re-activated by revaccination up to 12 months post-second vaccination. The findings presented here do not contradict the view that revaccination of horses with Strangvac at an interval of 12 months, as is used to protect against equine influenza virus, is suitable [13].

There was a highly significant correlation between the IdeE-neutralising effect of sera from vaccinated ponies with the level of IdeE-specific Ab, providing evidence to support the role of the antibody response in neutralising IdeE. Since pre-V3 IdeE-neutralisation values and IdeE-specific Ab values correlate, IdeE-specific Ab titres measured by ELISA could be considered as an indirect marker of antibody functionality (i.e., neutralisation). This correlation also indicates that the antibodies directed against IdeE generated after Strangvac vaccination are consistently active between individual animals and experiments.

As one major limitation of the study is the low number of animals in the groups, future studies with larger numbers of horses are required to address important remaining questions such as if the duration of protection and IdeE inhibition extends beyond 12 months or if there is an effect of age, or sex, on the functionality of anti-IdeE antibodies generated after vaccination. It was observed that the antibodies titres against the vaccine antigens did not return to the basal pre-vaccination levels up to one year post-second vaccination, but a clear cutoff titre that correlated with protection against *S. equi* was not identified. However, the IdeE-neutralisation titre could represent an additional datum to inform analyses of the duration of the protection.

Since it was observed that adding IdeE to the recombinant vaccine improves the protective efficacy [14,17] and IdeE is a secreted enzyme, it seems likely that the protective contribution is due to the neutralisation of the activity of this enzyme. Our data emphasizes the importance of focusing not only on those antigens on the bacterial surface that can be used in vaccines to generate opsonizing antibodies but also secreted virulence factors and toxins. A bacterin vaccine based on *S. equi* would be expected to contain bacterial surface proteins but might fail to present IdeE as an antigen. Therefore, the inclusion of recombinant IdeE in Strangvac provides one explanation for the excellent levels of protection afforded by this vaccine.

Finally, the mechanism of protection afforded by the other seven recombinant proteins, which are produced as two fusion proteins, CCE and Eq85, remains to be investigated. All these remaining antigens are bacterial surface proteins that are mostly involved in the attachment of the bacterium to host tissues. Antibodies that not only opsonize the bacteria but also inhibit the function of these adhesins would potentially counteract the attachment of the pathogen to the host tissues, offering a protective effect during the initial phase of the infection by reducing the number of *S. equi* that are able to attach to the equine mucosa.

## 5. Conclusions

In conclusion, a significant IdeE-specific antibody response in blood serum collected from ponies was induced after vaccination with Strangvac. Concomitantly, significant increases in the neutralising activity of IdeE occurred, persisting for at least 12 months post-second vaccination. IdeE-neutralising activity was increased significantly post-third vaccination 12 months after the second dose, demonstrating that the immunological memory response to the vaccine persisted for 12 months. Therefore, the administration of booster vaccinations every twelve months is expected to maintain an IdeE-neutralising effect.

## Figures and Tables

**Figure 1 vaccines-13-01061-f001:**
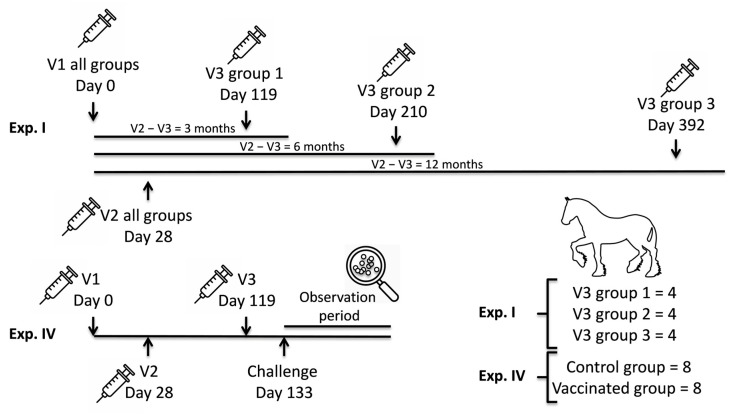
Immunization schedule. Ponies received first (V1), second (V2) and third (V3) vaccinations in Experiment (Exp.) I (*n* = 12) and IV (*n* = 8 vaccinated and *n* = 8 controls). The time of vaccination (Exp. I and IV) and experimental challenge infection with *S. equi* two weeks after the third vaccination (Exp. IV) are indicated.

**Figure 2 vaccines-13-01061-f002:**
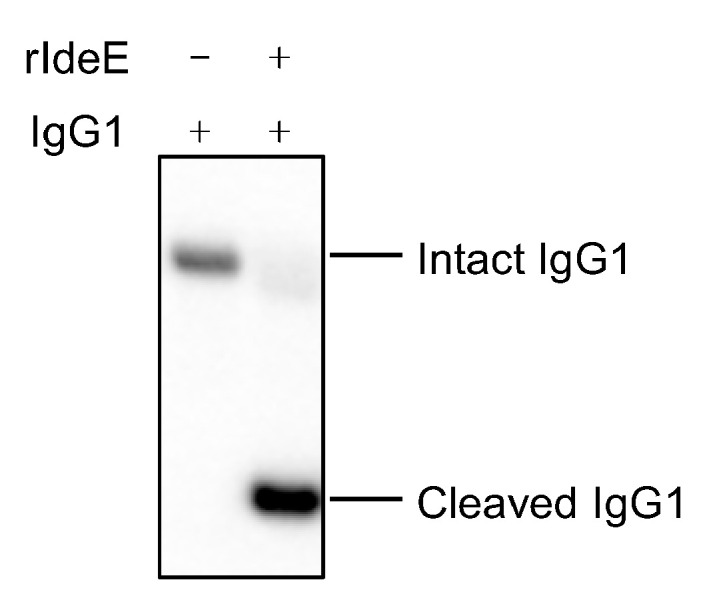
Western blot showing intact and cleaved IgG1 with and without treatment with recombinant IdeE (rIdeE).

**Figure 3 vaccines-13-01061-f003:**
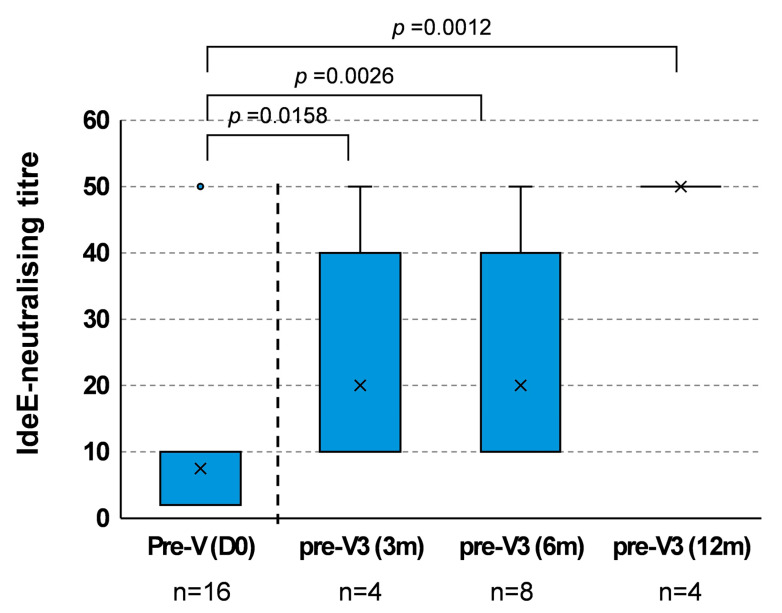
Duration of IdeE-neutralising activity after V2 and prior to V3. Serum pre-first vaccination (pre-V (D0)) and pre-third vaccinations (pre-V3 (3 m) = d119; pre-V3 (6 m) = d210 and pre-V3 (12 m) = d392) were used to measure IdeE-neutralising activity. Box and whisker plot, with whiskers show variation outside the upper and lower quartiles. The line and cross indicate median and mean values, respectively. Circles indicate outliers. n = number of samples; 4 samples taken from group 3 horses at the time of group 2 pre-V3 were included in the analysis (pre-V3 (6 m) n = 8 with their respective pre-V (D0); cf Appendix A and Appendix A). Significance set at *p* ≤ 0.016 after Bonferroni correction.

**Figure 4 vaccines-13-01061-f004:**
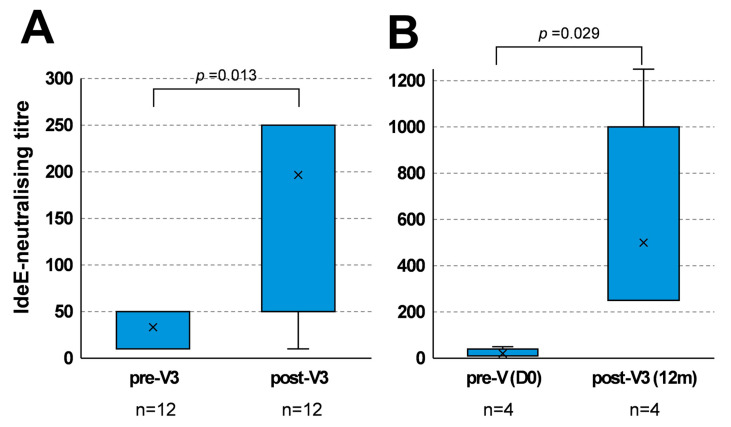
IdeE-neutralising activity after V3. (**A**) IdeE-neutralising activity immediately before V3 (pre-V3) were compared with activity measured 2 weeks after V3 (post-V3), irrespective of the study group. One outlier in the post-V3 group is located beyond the scale range. (**B**) IdeE-neutralising activity induced by V3 administered at the longest V2–V3 interval (12 m; post-V3 (12 m)) was compared with pre-vaccination IdeE-neutralising activity (pre-V d0). Box and whisker plot, with whiskers show variation outside the upper and lower quartiles. The line and cross indicate median and mean values, respectively. n = number of samples. Significance set at *p* ≤ 0.05.

**Figure 5 vaccines-13-01061-f005:**
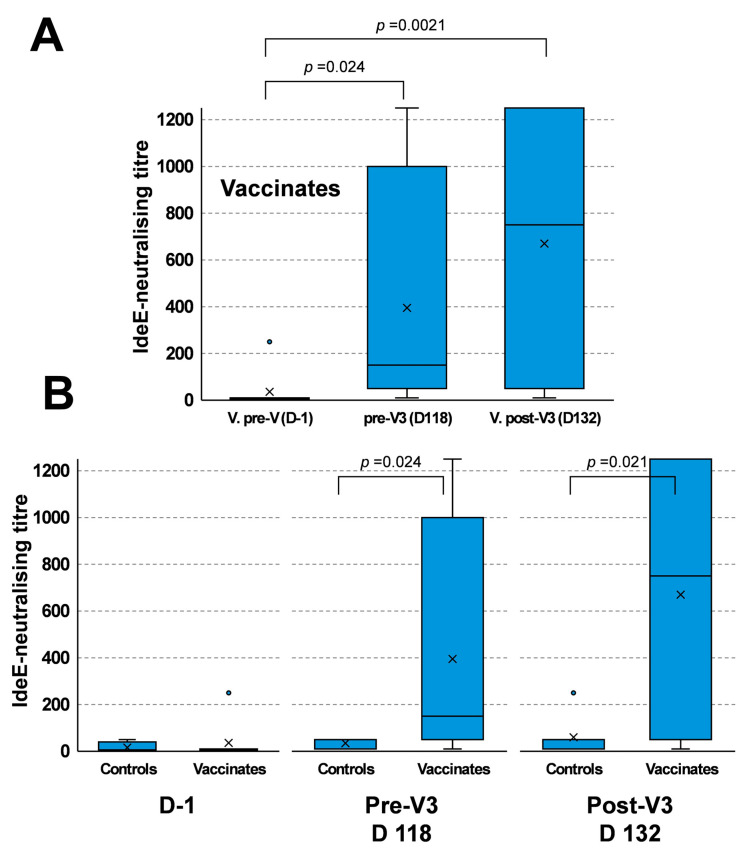
Neutralisation of IdeE activity in Experiment IV (Control placebo vv Vaccinates). (**A**) IdeE-neutralising activity in the vaccinated group. (**B**) Comparison of IdeE-neutralising activity between the placebo group (Controls) and vaccinated group (Vaccinates) at day 1, day 118 (prior to V3) or day 132 (post-V3). Box and whisker plot, with whiskers show variation outside the upper and lower quartiles. The line and cross indicate median and mean values, respectively. Circles indicate outliers. Significance set at *p* ≤ 0.05.

**Figure 6 vaccines-13-01061-f006:**
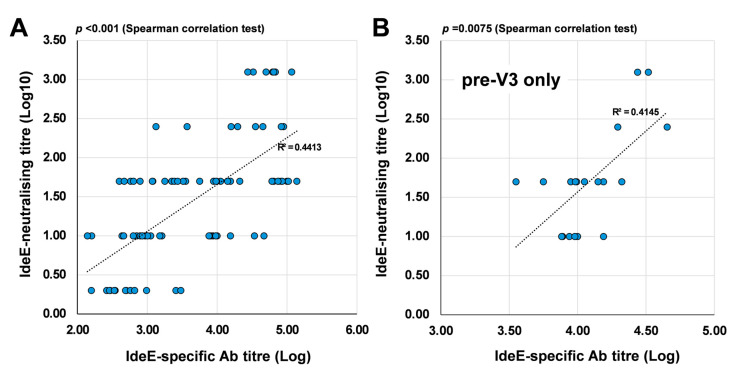
Correlation between IdeE-neutralising responses and the titre of IdeE-specific antibodies in sera. (**A**) Overall correlation analysis with all-time points and group serum samples (n = 84) included. (**B**) Correlation at pre-V3 only (n = 20). Both titres are log transformed for presentation purposes. The data used in this analysis are collected in Appendix A. R^2^ and linear regression curve are presented. Significance set at *p* ≤ 0.05.

## Data Availability

The original contributions presented in this study are included in the article/Appendix A. Further inquiries can be directed to the corresponding author.

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
