# Peer review of "Neutralisation of the Immunoglobulin-Cleaving Activity of Streptococcus equi Subspecies equi IdeE by Blood Sera from Ponies Vaccinated with a Multicomponent Protein Vaccine"

_vaccines, 2025, doi:10.3390/vaccines13101061_

Round 1
Reviewer 1 Report
Comments and Suggestions for Authors
This study provides valuable insights into the immune response to IdeE following Strangvac vaccination, particularly regarding neutralising activity and immune memory. However, addressing the academic questions above—especially the link between neutralisation and clinical protection, and the relevance of human IgG1 as a substrate—will strengthen the manuscript’s biological and translational significance. Correcting the writing errors (e.g., ambiguous pronouns, statistical terminology, formatting inconsistencies) will enhance clarity and align the work with the rigorous standards of Vaccines. Overall, the study advances understanding of Strangvac’s mechanism but requires additional context to confirm IdeE’s role in protective immunity.
The study employs human IgG1 as the substrate for measuring IdeE’s proteolytic activity and neutralisation (Sections 2.3, 3.1), yet IdeE is a virulence factor of Streptococcus equi (S. equi) that infects equids, not humans. Previous work indicates IdeE cleaves equine IgG (e.g., [15, 26]), and IdeE2—its paralog—exhibits species-specific cleavage efficiency for equine IgG [16]. Could the authors address:
- Why human IgG1 was selected over equine IgG as the substrate, especially given that vaccine-induced antibodies are intended to neutralise IdeE’s activity against equine immunoglobulins in vivo?
- Whether pilot experiments confirmed that human IgG1 cleavage by IdeE correlates with equine IgG cleavage (e.g., comparable cleavage kinetics or neutralisation titres when using both substrates)? Without this validation, the biological relevance of the neutralisation data to S. equi infection in horses remains unclear.
The manuscript demonstrates that Strangvac induces IdeE-specific antibodies and neutralising activity but does not establish a direct causal link between this neutralisation and protection against clinical strangles. Experiment IV included a challenge with S. equi (Section 2.1), yet the authors do not correlate individual IdeE neutralisation titres with clinical outcomes (e.g., abscess formation, nasal discharge).
- Present stratified data showing whether ponies with higher post-vaccination IdeE neutralisation titres (e.g., ≥1:750) had lower clinical scores or higher protection rates than those with lower titres?
- Discuss why no correlation was explored, especially given that prior studies linked IdeE to impaired equine neutrophil function [26]—a key defense against S. equi. Without this, the manuscript only describes an immune response, not a protective one.
The study reports a significant correlation between IdeE-specific antibody titres (iELISA) and neutralisation activity (Section 3.5) but does not confirm that the iELISA detects neutralising antibodies specifically (vs. non-functional binding antibodies). Since iELISAs often measure total binding antibodies (which may not inhibit enzyme activity).
- Whether the iELISA antigen (recombinant IdeE) includes the active site of the enzyme, and if so, whether pre-incubating sera with catalytically inactive IdeE abolishes the correlation (to confirm binding to functional epitopes)?
- Whether sera from naturally infected (non-vaccinated) horses—known to develop anti-IdeE antibodies [26]—exhibit similar neutralisation:ELISA correlations, to validate the assay’s relevance beyond vaccination?
The manuscript shows IdeE neutralising activity persists for 12 months post-V2 (Section 3.2) but provides no data on longevity beyond this timepoint. Given that equine vaccines often require booster intervals >12 months for field use, and the study recommends annual boosters (Section 5).
- Discuss whether preliminary data or literature precedents support the stability of neutralising activity (or immune memory) beyond 12 months? For example, do IdeE-specific memory B cells persist in ponies >12 months post-V2, and if so, what is their frequency?
- Explain why a 12-month booster interval is recommended over longer intervals (e.g., 18–24 months), especially if neutralising titres remain above protective thresholds (yet to be defined) at 12 months?
The study includes ponies of varying ages (4–5 months vs. 10–26 months) and sexes (e.g., Exp. I: 2 males, 10 females; Exp. IV: 10 males, 6 females; Section 2.1) but does not analyse whether these variables influence neutralising responses. Equine immune responses are known to vary with age (foals vs. yearlings) and sex [e.g., 31].
- Clarify whether age/sex stratification revealed differences in IdeE neutralisation titres (e.g., higher titres in older vs. younger ponies)?
- Explain why these variables were not controlled for in statistical analyses, especially if they could confound interpretations of booster responses or longevity data?
Section 1, Line 12: "These outwardly healthy ‘carriers’ intermittently shed S. equi into the environment where the organism can be taken up by naïve animals, triggering new outbreaks of disease [5–7]." "The organism" is ambiguous (could refer to "S. equi" or "abscess material").
Revision: "These outwardly healthy ‘carriers’ intermittently shed S. equi into the environment, where the bacterium can be taken up by naïve animals, triggering new outbreaks of disease [5–7]."
Section 2.5, Line 2: "Data were found to be not normally distributed and the non-parametric Friedman post-hoc test (for repeated measures) and the Kruskal-Wallis post-hoc test (for independent measures) were used..."
"Friedman" and "Kruskal-Wallis" are primary tests, not "post-hoc" tests. Post-hoc tests (e.g., Nemeney for Friedman) are used after primary tests to identify pairwise differences.
Revision: "Data were found to be non-normally distributed; the non-parametric Friedman test (for repeated measures) and Kruskal-Wallis test (for independent measures) were used as primary analyses. Post-hoc pairwise comparisons were performed using the Nemeney test (for Friedman) and Bonferroni-corrected tests (for Kruskal-Wallis), with significance set at P<0.05."
Section 4, Line 1: "Approximately 75% of horses that recovered from natural strangles developed immunity to S. equi infection that, shortly after recovery, provided resistance to a 10-fold greater challenge inoculum than that to which they were originally susceptible [1,20]."
"Immunity" and "resistance" are redundant in this context.
Revision: "Approximately 75% of horses that recovered from natural strangles developed immunity to S. equi, providing resistance shortly after recovery to a 10-fold greater challenge inoculum than that to which they were originally susceptible [1,20]."
Author Response
We thank reviewer #1 for taking the time to review this manuscript. Please find the detailed responses in the attachment and the corresponding revisions/corrections highlighted in the re-submitted manuscript file.

Reviewer 2 Report
Comments and Suggestions for Authors
This paper by Righetti et al. provides data on the antibodies produced to the immunoglobulin cleaving endopeptidase IdeE following vaccination with the multi-antigen fusion protein (Strangvac, Intervacc AB) administered with the adjuvant Matrix V of Novavax AB.
The functional antibody tests that neutralised the immunoglobulin cleavage activity of IdeE showed that the titres could be persistent and importantly that they could be boosted 12 months after the last injection. The neutralising test was correlated with IgG antibodies measure to the IdeE antigen by ELISA.
This is not a protection assay but the paper proffers a strong and interesting argument that anti-IdeE neutralising antibodies are a major determinant of the protective immunity. The authors have produced some of the leading papers in the development of the vaccine against S. equi and the assays presented in the paper and in the supplementary section appear to be well executed so the information should be valuable for veterinarians and horse keepers as well as to researcher who are able to work further on protection assays.
Did the study use human immunoglobulin for the target of the IdeE endopeptidase as written in the paper? It has been shown produce species-specific cleavage so a neutralising assay against the natural target would be more convincing especially if the affinity of binding is important for competition.
It is not clear from tracking back on the method or production if the Strangvac was insoluble or aggregated before being absorbed onto the Matrix V nanoparticles. This could be significant for quality control so some information would be relevant.
Author Response
We thank reviewer #2 for taking the time to review this manuscript. Please find the detailed responses in the attachment and the corresponding revisions/corrections highlighted in the re-submitted manuscript file.

Reviewer 3 Report
Comments and Suggestions for Authors
The study is well conducted and very promising. As a suggestion, it would be interesting in the future to evaluate this experimental vaccine alongside a commercial bacterin vaccine.
Also, as mentioned by the author, it will be very important to evaluate the role of the different antigens in the immune response and protection. This point seems more important considering the different amounts of antigens used in the vaccine formulation.
Author Response
We thank reviewer #3 for taking the time to review this manuscript. Please find the detailed responses in the attachment and the corresponding revisions/corrections highlighted in the re-submitted manuscript file.
